# Genome-Wide Analysis of the *GLK* Gene Family and Its Expression at Different Leaf Ages in the *Citrus* Cultivar Kanpei

**DOI:** 10.3390/plants13070936

**Published:** 2024-03-23

**Authors:** Bo Xiong, Hongzhen Chen, Qingqing Ma, Junfei Yao, Jialu Wang, Wenjia Wu, Ling Liao, Xun Wang, Mingfei Zhang, Siya He, Jiaxian He, Guochao Sun, Zhihui Wang

**Affiliations:** College of Horticulture, Sichuan Agricultural University, Chengdu 611130, China; xiongbo1221@sicau.edu.cn (B.X.); 15775554829@163.com (H.C.); 202201646@stu.sicau.edu.cn (Q.M.); 19845921559@163.com (J.Y.); wangjial0321@163.com (J.W.); 202302925@stu.sicau.edu.cn (W.W.); liaoling0102@sicau.edu.cn (L.L.); wx0104@sicau.edu.cn (X.W.); zhang_mingfei@sicau.edu.cn (M.Z.); hesiya1114@163.com (S.H.); hejx1231@163.com (J.H.); sunguochao@163.com (G.S.)

**Keywords:** *Citrus*, *GLK*, leaf age, photosynthetic characteristics, chlorophyll

## Abstract

The *GLK* gene family plays a crucial role in the regulation of chloroplast development and participates in chlorophyll synthesis. However, the precise mechanism by which *GLK* contributes to citrus’s chlorophyll synthesis remains elusive. The *GLK* gene family causes variations in the photosynthetic capacity and chlorophyll synthesis of different citrus varieties. In this study, we identified tissue-specific members and the key *CcGLKs* involved in chlorophyll synthesis. A total of thirty *CcGLK* transcription factors (TFs) were discovered in the citrus genome, distributed across all nine chromosomes. The low occurrence of gene tandem duplication events and intronic variability suggests that intronic variation may be the primary mode of evolution for *CcGLK* TFs. Tissue-specific expression patterns were observed for various *GLK* family members; for instance, *CcGLK12* and *CcGLK15* were specifically expressed in the skin, while *CcGLK30* was specific to the ovary, and *CcGLK10*, *CcGLK6*, *CcGLK21*, *CcGLK2*, *CcGLK18*, *CcGLK9*, *CcGLK28*, and *CcGLK8* were specifically expressed in the leaves. *CcGLK4*, *CcGLK5*, *CcGLK11*, *CcGLK23*, *CcGLKl7*, *CcGLK26*, and *CcGLK20* may participate in the regulation of the ALA, prochlorophylate, protoporphyrin IX, Mg-protoporphyrin IX, Chl *b*, T-Chl, MG-ProtoIX ME, and POR contents in citrus.

## 1. Introduction

Studies have shown that photosynthetic capacity and fruit quality are influenced by the transcription factor *GLK* (*Golden 2-like*), which exerts regulatory control over chloroplast development and chlorophyll synthesis [1]. The *GLK* transcription factor regulates chloroplast development and the formation of photosynthetic organs in *Arabidopsis* [2]. Previous studies have reported that *GLK* family transcription factors play a crucial role in regulating chloroplast development and chlorophyll accumulation in rice, peach, and other plants [3,4,5]. Additionally, *GLK* is involved in modulating various activities of the chlorophyll synthases, including δ-aminoacetobulinate dehydratase (ALAD), porphyrinogen deaminase (PGBD), protoporphyrinogen oxidase (PROTOX), etc. [6]. The induction of *GLK* leads to an upregulation in the production of diethylene protochlorophylates, chlorophyll a (Chl *a*), and chlorophyll b (Chl *b*) [7].

An overexpression of the *GLK* gene in tomato (Solanum lycopersicum) can increase its plastid number and pigment content [8]; *GLK* is essential for chloroplast development [9], while *ZmGLK1* and *ZmGLK2* regulate the transition from plastids to chloroplasts in maize’s vascular bundle sheath cells and mesophyll cells, respectively [10,11]. A deficiency in *GLK* leads to disturbed leaf chloroplast development [12], but an overexpression of *GLK* enhances chloroplast biogenesis and photosynthesis even in nonphotosynthetic organs such as roots and fruits [1,13]. Moreover, *GLK* plays a role in regulating multiple activities related to chlorophyll synthesis genes including δ-aminoacetobulinate dehydratase (ALAD), porphyrinogen deaminase (PGBD), protoporphyrinogen oxidase (PROTOX), magnesium chelatase (MGCH), magnesium protoporphyrin cyclase (MPECYC), and proprodol oxidoreductase (POR), which are regulated by the transcription factors *MYB7* and *GLK* [5,6].

## 2. Results and Analysis

### 2.1. Chlorophyll Synthesis Precursor Content and Precursor Synthase Activity

The levels of the key precursor for chlorophyll synthesis were determined, revealing elevated concentrations of biledochromatogen, uroporphyrinogen III, fecal porphyrinogen III, Chl a, Chl b, and T-Chl. Additionally, variations in the growth period of different precursors were observed. The results showed that ALA, uroporphyrin III, Chl a, Chl b, and T-Chl exhibited rapid increases from D3 to D27, followed by slower increments from D37 to D47. PBG, fecal porphyrinogen III prochlorate protoporphyrin IX, and Mg-protoporphyrin IX showed continuous or rapid increases from D27 to D47 (Figure 1).

The activity of glutamyl-tRNA reductase (Glu-tRNA) exhibited a rapid increase from D3 to D27, reaching a plateau thereafter. Conversely, the activities of uroporphyrinogen decarboxylase (UROD) and magnesium protoporphyrin IX methyltransferase (ChlM) showed a rapid increase from D27 to D47. However, the content of magnesium protoporphyrin IX monomethyl ester cyclase (Mg-ProtoIX ME) remained relatively unchanged (Figure 2).

### 2.2. Expression Levels of Key Genes for Chlorophyll Synthesis

The expression levels of *CAO*, *CHLD*, *PORA*, *PORB*, and *PORC* exhibited a continuous increase. Notably, *CHLM* played a regulatory role in the synthesis of magnesium protoporphyrin IX mono-methyl ester cyclase (Mg-ProtoIX ME), which was consistent with the observed changes in enzyme content and gene expression during each time period. Specifically, *CHLH* expression showed a rapid increase from D17 to D27, *GUN4* expression increased from D3 to D27, and *HEMA1* expression increased from D17 to D37 (Figure 3).

### 2.3. Identification and Basic Physical and Chemical Properties of the CcGLK Proteins

A total of 30 eligible GLK protein sequences, designated *CcGLK1*~*CcGLK30*, were obtained. Through motif composition and phylogenetic analysis, the 30 *CcGLKs* were classified into three subfamilies. Additionally, their fundamental physicochemical properties were predicted and analyzed (Table 1). The transcription factors exhibited amino acid lengths ranging from 122 to 496, molecular weights ranging from 14,435.59 to 54,724.06, isoelectric points ranging from 5.57 to 10.46, instability coefficients ranging from 27.46 to 70.95, and hydrophobicity indices varying between −1.057 and −0.246.

### 2.4. Phylogeny, Conserved Motifs, and Gene Structure of CcGLKs

Based on its evolutionary relationships and a motif analysis, the *GLK* family is classified into three distinct groups. Subfamily I comprised 13 members, while subfamilies II and III consisted of 10 and 7 members, respectively (Figure 4).

The conserved motifs of the 30 CcGLK proteins were analyzed using the online tool MEME, resulting in a total of 20 identified motifs (Figure 5A,B). The functional annotation of these motifs was performed using the CDD tool, revealing five putative functional annotations, namely Myb-CC-LHEQLE, Myb DNA binding, the Myb-CC-LHEQLE superfamily, the SANT superfamily, and Myb SHAQKYF. Notably, all *CcGLKs* exhibit structural conservation, with the presence of Myb DBD, Myb SHAQKYF, and the SANT superfamily throughout.

In the analysis of gene structure, a distinct distribution of intron regions was observed in the *CcGLK* gene (ranging from 0 to 8), and, generally, the *CcGLKs* that clustered together exhibited comparable exon/intron architectures (Figure 5D), encompassing both their intron count and exon length. Alterations in the intron count may represent a pivotal factor contributing to evolutionary diversification in terms of the gene’s structure and functionality.

To further assess the similarity between the citrus *GLK* domains, we aligned the 30 *CcGLKs*’ domain sequences using GeneDoc (Figure 6). The results revealed that the Myb DBD in *CcGLKs* contained an HLH structure with two highly conserved regions. The first helix consistently contained a 14-amino acid sequence PELHRR, while the second helix features an NI/VASHLQ motif at its beginning. These helices were separated by loops consisting of 22 amino acids.

### 2.5. Chromosome Position, Collinearity Analysis, and Gene Duplication of CcGLKs

The distribution of *GLK* genes was non-uniform across chromosomes (Chr), with chromosome 2 harboring the highest number of *CcGLK* TFs (6), while only one *GLK* gene was present on chromosome 1 (Figure 7).

The gene duplication events in the *Citrus* genome were estimated through collinearity analysis, revealing that only two tandem duplication events occurred during their evolution. These events involved four genes (*CcGLK*15, *CcGLK*18, *CcGLK*28, *CcGLK*9) located on Chr 2, Chr 4, and Chr 6 (Figure 8A). The Ka/Ks values of the collinearity pairs were less than 1 (Table 2), indicating that the evolutionary trajectory of the *CcGLK* TFs has not been strongly influenced by purifying selection.

To further investigate the evolutionary mechanism of citrus *CcGLKs*, we constructed a comparative collinear map of *Citrus* with two model plants: *Arabidopsis* and *rice*. The results revealed that there were 30 homologs between *Citrus* and *Arabidopsis* (Table 3), as well as 18 between *Citrus* and *rice* (Table 4). Notably, 10 *CcGLKs* exhibited collinearity with both *Arabidopsis* and *rice*, suggesting their existence predates the divergence of monocots and dicots. Furthermore, the number of *CcGLK*-*OsGLK* pairs was found to be lower than that of the *CcGLK*-*AtGLK* pairs, indicating an earlier divergence between the common ancestor of rice and dicots compared to the divergence between *Citrus* and *Arabidopsis*.

### 2.6. Citrus GLK Expression Pattern

In order to investigate the tissue-specific expression patterns of the *GLK* gene family members, we generated expression profiles based on transcriptome sequencing data from five different tissues. Subsequently, a clustering analysis was performed to identify members with similar expression patterns (Figure 8). The results showed that *CcGLK*4, *CcGLK*20, *CcGLK*25, *CcGLK*16, *CcGLK*17, *CcGLK*7, *CcGLK*5, *CcGLK*11, and *CcGLK*26 exhibited relatively high expression levels in all four tissues [log_2_(FKPM + 1) > 3]. Conversely, low expression levels were observed for *CcGLK*30, *CcGLK*19, *CcGLK*18, *CcGLK*15, *CcGLK*9, *CcGLK*28, *CcGLK*24, and *CcGLK*29 across all four tissues [log_2_(FKPM + 1) < 1]. Notably, the pericarp specifically expressed *CcGKL12* and *CcGKL15*, the ovary specifically expressed *CcGLK30*, and the leaves specifically expressed *CcGLK*10, *CcGLK*6, *CcGLK*21, *CcGLK*2, *CcGLK*18, *CcGLK*9, *CcGLK*28, and *CcGLK*8.

The highly expressed *GLK* members [log_2_(FPKM + 1) ≥ 5] were screened out to analyze whether they have similar expression patterns in the variety ‘Kanpei’. The results are shown in Figure 9; the highly expressed *GLK* members in *Clementine* also had high expression levels in ‘Kanpei’, and, at leaf maturity (D47), *CcGLK2*, *CcGLK5*, *CcGLK11*, *CcGLK21*, *CcGLK23*, and *CcGLK25* all had relatively high expression levels.

Meanwhile, the expression of the *CitGLKs* exhibited temporal fluctuations across different growth stages. The expression levels of *CcGLK2*, *CcGLK5*, *CcGLK11*, *CcGLK16*, *CcGLK20*, *CcGLK21*, and *CcGLK25* displayed oscillations during growth. However, they consistently demonstrated higher expression levels. Conversely, the expression levels of *CitGLK4* and *CitGLK17* were elevated from D3 to D17 but declined from D37 to D47 (Figure 10).

### 2.7. Correlation Analysis

The correlation analysis between the *CcGLKs* and chlorophyll precursors, as well as precursor synthase, revealed that *CcGLK4* exhibited a negative correlation with protochlorophylls, protoporphyrin IX, and Mg-protoporphyrin IX. On the other hand, *CcGLK5* and *CcGLK11* displayed positive correlations with protochlorophyllin, protoporphyrin IX, and Mg-protoporphyrin IX. Additionally, both *CcGLK11* and *CcGLK23* showed positive correlations with the leaves’ Chl b and T-Chl content. Moreover, there was a positive correlation observed between *CcGLK26* and ALA content. Furthermore, the content of Mg-ProtoIX ME demonstrated a positive correlation with *CcGLK17* while the POR content exhibited a positive correlation with *CcGLK20* (Figure 11).

## 3. Materials and Methods

### 3.1. Test Materials

In this experiment, ‘Kanpei’ {[(*C. unshiu × C. sinensis*) *× C. sinensis*] *× C. reticulata*} was used as the material, which was grafted on Ziyang xiangcheng. The growth conditions were optimal, with no presence of diseases or insect pests. Cultivation management and growth practices remained consistent throughout the study period. Sampling commenced on the third day after summer buds began to emerge, with whole shoots being collected every seven days during the early stage of budding, specifically targeting the second to third leaves from the base of each branch. During the middle and late stages of summer bud emergence, sampling was conducted every ten days. A total of six sampling time points were included in this study. The first 2–3 leaves were collected from the base of the branch, washed with tap water, moistened with deionized water, and subsequently dried (Figure 12). After thorough mixing, the sample was randomly divided into two portions. One portion was utilized for chlorophyll and precursor content determination, while the other part was rapidly frozen using liquid nitrogen and stored at −80 °C in a refrigerator for subsequent analysis of its enzyme activity and gene expression.

### 3.2. GLK Gene Family Members and Prediction of Their Physicochemical Properties

*Arabidopsis* gene family protein sequences were downloaded from Plant TFDB V5.0 (http://planttfdb.gao-lab.org/index.php, accessed on 15 April 2022). Hidden Markov models (HMMs) were constructed using HMMER 3.0 software (http://hmmer.org/, accessed on 17 April 2022), and HMMER 3.0 software was employed to search for GLK family protein sequences in the citrus proteome (http://citrus.hzau.edu.cn/orange/index.php, version2, accessed on 17 April 2022). The CD Search website was utilized to examine the domains of all candidate GLK proteins in citrus. Finally, only those sequences with an intact *MYB* DNA-binding domain were retained and designated as *CcGLK*1~30 in *Citrus clementina*.

### 3.3. Chromosomal Distribution, Gene Duplication, and Collinearity Analysis of CcGLKs

The chromosomal distribution of *GLK* genes was retrieved from gff3 files in the *Citrus* pan-genome breeding database (http://citrus.hzau.edu.cn/index.php, accessed on 20 April 2022) and visualized using Map Inspect. Gene duplications in *CcGLK* TFs were analyzed using MC Scan X with default parameters. Additionally, collinearity profiles between citrus, *Arabidopsis*, and rice were constructed using TBtools software 2022.

### 3.4. Determination of Chlorophyll Synthesis Precursor Content and Key Precursor Synthases’ Activity

The chlorophyll content was determined using the method described by Moran and Porath [14]. A 0.5 g sample of the leaves was cut and placed in a 15 mL centrifuge tube, followed by the addition of 10 mL of 95% ethanol (*v*/*v*) and 5 mL of an 80% acetone solution. The mixture was kept in the dark for 24 h until the leaves turned white. OD values of 663 nm, 645 nm, and 470 nm were measured using an enzyme calibration system (Thermo Fisher Scientific, multiskan go, Waltham, MA, USA). We calculated the concentration of each pigment in the leaves (mg·g^−1^) according to the following equations:Chl *a* content = 12.21 × OD663 − 2.81 × OD645.
Chl *b* content = 20.13 × OD645 − 5.03 × OD663.

Carotenoid content = (1000 × OD474 − 3.27a − 104b)/229 (a and b indicate the content of Chl *a* and Chl *b*, respectively).
T-Chl content = Chl *a* + Chl *b*.

The determination of Proto-IX, Mg-Proto IX, and Pchlide content was performed as described by Liu et al. [15]. A 0.3 g leaf sample was weighed, appropriate liquid nitrogen was added to it, and then it was ground. Then, 10 mL extraction solutions were added (acetone/ammonia water = 9:1), homogenized thoroughly, and then centrifuged at 12,000 rpm for 10 min. The OD of the supernatant was measured at 575 nm, 590 nm, and 628 nm, respectively. The content was calculated using the following equations:Proto-IX content = 0.18016 × OD575 − 0.04036 × OD628 − 0.04515 × OD590.
Mg-Proto IX content = 0.06077 × OD590 − 0.01937 × OD575 − 0.003423 × OD628.
Pchlide content = 0.03563 × OD628 + 0.007225 × OD590 − 0.02955× OD575.

A Plant Glu-tRNAs ELISA Kit (catalog number: ZK-8377), plant Magnesium protoporphyrin IX methyltransferase (ChlM) ELISA Kit (catalog number: ZK-8381), and plant uroporphyrinogen decarboxylase (UROD) ELISA Kit (catalog number: ZK-8383) (Shanghai Zhen Ke Biological Technology Co., Ltd., Shanghai, China) were used to extract enzymes and determine the activity of key precursor synthases.

### 3.5. Characterization, Phylogenetic, and Physicochemical Properties Analysis of CcGLKs

MEGA was used to construct a phylogenetic tree based on the neighbor joining (N–J) method, with a bootstrap value of 1000. The tree was beautified using the Evolview website (http://120.202.110.254:8220/evolview, accessed on 16 April 2022). The specific conserved domains in the MAPK family were searched on NCBI (https://www.ncbi.nlm.nih.gov, accessed on 1 July 2021) and the conserved domains were predicted by InterProScan (http://www.ebi.ac.uk/Tools/InterProScan/1, accessed on 1 July 2021). MEME was used to identify conserved motifs and TB tools software was used to visualize the evolution motifs and e genetic structures of the *GLK* family in cultivated strawberry. Subcellular localization analysis was performed using WOLF PSORT (https://www.genscript.com/wolf-psort.html/, accessed on 5 July 2021).

### 3.6. The Analysis of the Expression Patterns of CcGLKs

Based on a transcriptomic data analysis, the expression of *CcGLKs* in various tissues (ovary, fruit, peel, pulp, leaf) was assessed and quantified as fragments per million kilobase (FPKM). To facilitate hierarchical clustering log_2_(FPKM + 1) was employed. The results were visualized using TBtools.

The expression changes of *GLK* members were analyzed by quantifying the *GLK* members in leaves (log_2_(FPKM + 1) ≥ 5) under different growth periods of ‘Kanpei’. *CcGLK29*, which exhibited the lowest expression in Period 1 (D3), was selected as the reference gene for normalization. The relative gene expression values were calculated using the 2^−∆∆Ct^ method, and the primer sequences are listed in Table 5.

### 3.7. Data Processing and Analysis

A significance analysis was performed using SPSS 23.0 software (*p* < 0.05), and the results were visualized using OriginPro 2022 software.

## 4. Discussion

Leaves are an important source organ of plants, controlling the production of photosynthetic products for flowers, fruits, shoots, and other plant structures. Optimal leaf growth is crucial to ensure an adequate supply of photosynthetic products for plant development and to achieve high-quality and high-yield fruit production. This study investigates the photosynthetic characteristics, chlorophyll precursor content, key enzyme activity, and gene expression in ‘Kanpei’ leaves while also identifying the expression patterns of the *GLK* genes across different tissues (ovary, fruit, peel, pulp, and leaf) and growth stages.

### 4.1. Differences in Key Precursor Content and Gene Expression of Chlorophyll Synthesis

Chl a and Chl b have key roles in the process of photosynthesis. Previous studies have demonstrated a positive correlation between the augmentation of photosynthetic activity and an increase in the content of Chl a and Chl b [16]. In this study, we found that a relatively low content of chlorophyll and its weak photosynthetic capacity may be caused by a low content of PBG, uroporphyrinogen III, fecal porphyrinogen III, and Mg-protoporphyrin IX.

At the molecular level, there is significant variation in the expression of the crucial genes involved in chlorophyll synthesis among different varieties. Glu-tRNA serves as a rate-limiting enzyme in the biosynthesis of tetrapyrrole precursor ALA, and its three *Arabidopsis* orthologs (*HEMA1*, *HEMA2*, and *HEMA3*) collectively encode isoforms of GluTR. The suppression of *HEMA1* through antisense expression leads to reduced levels of Chl and ALA in plants [17]. Therefore, HEMA1 is considered to play a major role in the biosynthesis of the tetrapyrrole precursor ALA. This study found that the Glu-tRNA in ‘Kanpei’ may promote the synthesis of ALA in ‘Kanpei’. *CHLH*, *GUN4*, and *CHLD* encode magnesium chelatase (MgCh), which is involved in the synthesis of the chlorophyll precursor Mg-protoporphyrin IX [18]. Previous studies have found that *POR* is involved in the biosynthesis of Chl a [18]. The expression levels of three *POR* homologs in ‘Kanpei’ exhibited distinct temporal patterns, potentially contributing to the observed high Chl a content. *CAO* is a gene catalyzing the conversion of Chl a to Chl b [19]. As leaf age increased, the expression of the *CAO* gene exhibited a gradual increase, consequently resulting in a growth rate of the Chl a content in the leaves that was lower than that of Chl b.

### 4.2. Identification and Expression Analysis of GLK Genes in Citrus

The *GLK*-conserved motif SHAQKYF, which is present in plants and yeast, functions as a transcriptional activator by binding to an I-box located in the DNA-binding domain at the C terminus [20]. In addition to two conserved binding domains, Group I contains a unique Myb-CC-LHEQLE domain within its Myb-CC region, representing potential functional diversity. This domain exhibits a highly conserved LHEQLE sequence that responds to various abiotic stresses such as phosphate starvation signals [21]. Members of the *GLK* gene family possess both the highly conserved HLH region (DNA-binding domain) and the GCT cassette (involved in dimerization) [11]. In many well-characterized transcription regulators, the HLH domain binds DNA and facilitates dimerization [22,23].

The DNA-binding domain sequences belong to the GARP family of transcription factors [24]. Additionally, both regions of the HLH DNA-binding domain exhibit conservation [25]. In this study, we identified both conserved regions of the HLH structure; however, it should be noted that the conserved sequences in citrus are not identical. Through multiple sequence alignment analysis, we observed a high degree of conservation in the second helix region (VK/VASHLQ) of the *CcGLK* gene. Conversely, various variants were found in the first helix region, with L and H being relatively conserved. These findings suggest that functional divergence within the citrus *GLK* gene family may primarily arise from variations in the first helix region. Similar observations regarding the functional diversity resulting from sequence variation have also been reported in maize studies [25].

Multigene families often arise from gene duplication, with amplification mechanisms including fragmentation/tandem duplication, retrotransposition, and genome polyploidization [26,27]. This study identified only two instances of gene duplication events (*CcGLK*15-*CcGLK*18, *CcGLK*28-*CcGLK*9) in the *GLK* family, suggesting that gene duplication may not be the primary driver of *GLK* family members’ evolution. The frequent loss and insertion of new introns play a crucial role in gene evolution. In eukaryotes, there is a significant reduction in their intron numbers during evolution [28], while an analysis of segmental duplication events in rice indicated more lost than acquired introns [29]. Our study revealed substantial variability in the intron distribution within the *CcGLK* gene, ranging from 0 to 8, implying that intron variation had been a key factor driving its evolutionary history since its origin. Gene structural variation was not solely reflected by changes in the exon number but also by alterations in exon length. These variations suggested that the elongation and termination of transcription could modify gene structure, lead to s gain or loss of domains, and potentially affect protein function.

According to classical theory, gene duplication results in two possibilities for each copy of a gene: the original features are retained by one copy through negative selection and the remaining copies become pseudogenes without selection [30]. *CcGLK*15, *CcGLK*18, and *CcGLK*28 exhibited little or no expression in any tissue or organ, suggesting that they are either pseudogenes or silent paralogs. Furthermore, most members of these genes belong to Group II.

The *GLK* gene plays a crucial role in plant chloroplast development. However, this study reveals that *GLK* also exhibited relatively high expression levels in non-photosynthetic organs. In the tissue-specific expression analysis of maize *GLK* genes (*ZmGLK2*, *ZmGLK9*, *ZmGLK28*, *ZmGLK35*, *ZmGLK44*) in roots, these genes may be implicated in stress responses [25]. Similarly, within the non-photosynthetic organ pulp, higher expression levels of *CcGLK*4, *CcGLK*5, *CcGLK7*, *CcGLK11*, *CcGLK20*, *CcGLK16*, and *CcGLK9* were observed, suggesting their potential involvement in regulating fruit’s development, maturity, or nutrient accumulation [1,31,32].

Several members of the *GLK* gene family exhibited significant expression levels across various leaf growth stages, including *CcGLK2*, *CcGLK5*, *CcGLK11*, *CcGLK16*, *CcGLK20*, *CcGLK21*, *CcGLK23*, and *CcGLK25*. These genes potentially play a crucial role in chlorophyll biosynthesis.

The induction of *GLK* has been demonstrated to upregulate three key processes [7]. Firstly, it promotes the generation of diethylene chlorophylate through a three-step reaction catalyzed by magnesium protoporphyrin IX monomeyl ester cyclase (CRD 1). Secondly, it enhances NADPH production for Chl a synthesis via DOLase (PORA, PORB, and PORC). Lastly, it facilitates the oxidation of chlorophyll A by chlorophyll A oxidase (CAO) to yield chlorophyll b. In this study, *CcGLK5*, *CcGLK11*, *CcGLK17*, *CcGLK20*, and *CcGLK23* exhibited significantly positive correlations with prochlorophylates, protoporphyrin IX, Mg-protoporphyrin IX, Chl b, and T-Chl contents, as well as the Mg-ProtoIX ME and POR enzymes. Previous research demonstrated that *GLK* induced a substantial upregulation of *HEMA1*, *CHLH*, *GUN4*, *CRD1*, and *CAO*. However, it did not affect other steps preceding protoporphyrin IX formation. Additionally, the expression levels of two other subunits of magnesium chelatase (*CHLD* and *CHLI*) remained unchanged. These findings contrast with those reported by Matsumoto and Stephenson [33,34], indicated that *GLK* regulated the activity of magnesium protoporphyrin IX through its transcriptional regulation of *CHLH* and *GUN4*. Similar results were also observed in our study, where 30% of the identified *CcGLK* members showed significant positive correlation with prochlorophylls, protoporphyrin IX, Mg-protoporphyrin IX, and Chl b.

## 5. Conclusions

In this study, we investigated the variations in the photosynthetic capacity and chlorophyll synthesis of ‘Kanpei’ at different leaf ages, while also exploring the citrus *GLK* gene family to identify the tissue-specific members and key *GLKs* involved in chlorophyll synthesis.

A total of 30 *CcGLK* transcription factors (TFs) were identified in the *Clementine* genome, which were further classified into three subgroups and distributed across nine chromosomes. The presence of the *GLK* gene family predates the divergence between monocots and dicots. The relatively low occurrence of gene tandem duplication events and intronic variability suggests that intronic variation may represent the primary mode of evolution for *CcGLK* TFs. The expression of *GLK* family members was tissue-specific, with *CcGLK*12 and *CcGLK*15 specifically expressed in the peel, *CcGLK*30 specifically expressed in the ovary, and *CcGLK*10, *CcGLK*6, *CcGLK*21, *CcGLK*2, *CcGLK*18, *CcGLK*9, *CcGLK*28, and *CcGLK*8 specifically expressed in the leaf. Correlation analysis between the expression of individual *CcGLKs* and chlorophyll precursors, and precursor synthases indicated that *CcGLK*4, *CcGLK*5, *CcGLK11*, *CcGLK23*, *CcGLK17*, *CcGLK26*, and *CcGLK*20 may be key members involved in the regulation of ALA, protoporphyrin IX, Mg-protoporphyrin IX, Chl b, T-Chl, Mg-ProtoIX ME, and POR synthesis.

## Figures and Tables

**Figure 1 plants-13-00936-f001:**
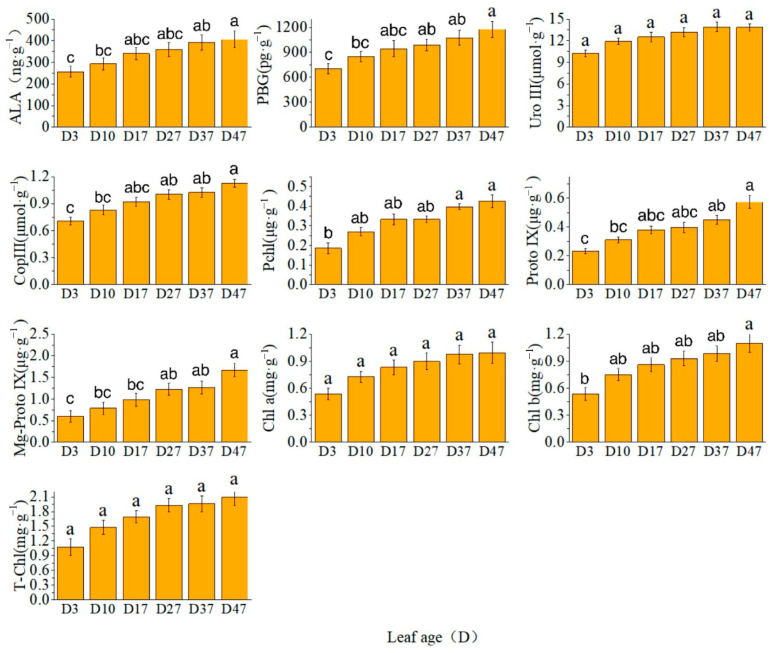
Content of chlorophyll and its precursors. Different letters above the bar chart indicate significant differences between different treatments (*p* < 0.05).

**Figure 2 plants-13-00936-f002:**
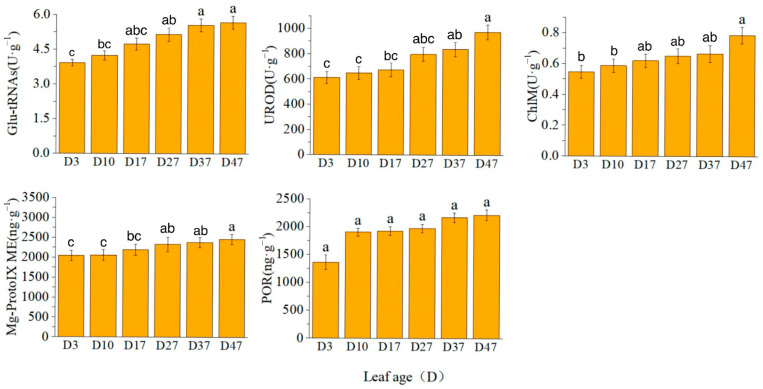
Chlorophyll synthase activity/content. Different letters above the bar chart indicate significant differences between different treatments (*p* < 0.05).

**Figure 3 plants-13-00936-f003:**
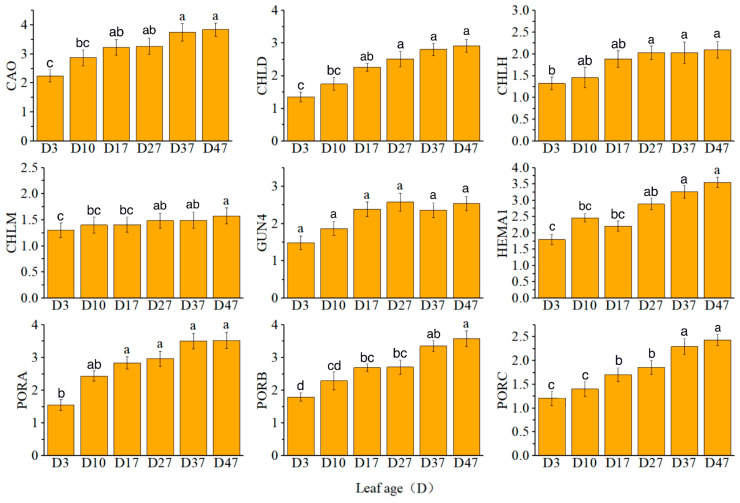
Relative expression levels of key genes in chlorophyll synthesis. Different letters above the bar chart indicate significant differences between different treatments (*p* < 0.05).

**Figure 4 plants-13-00936-f004:**
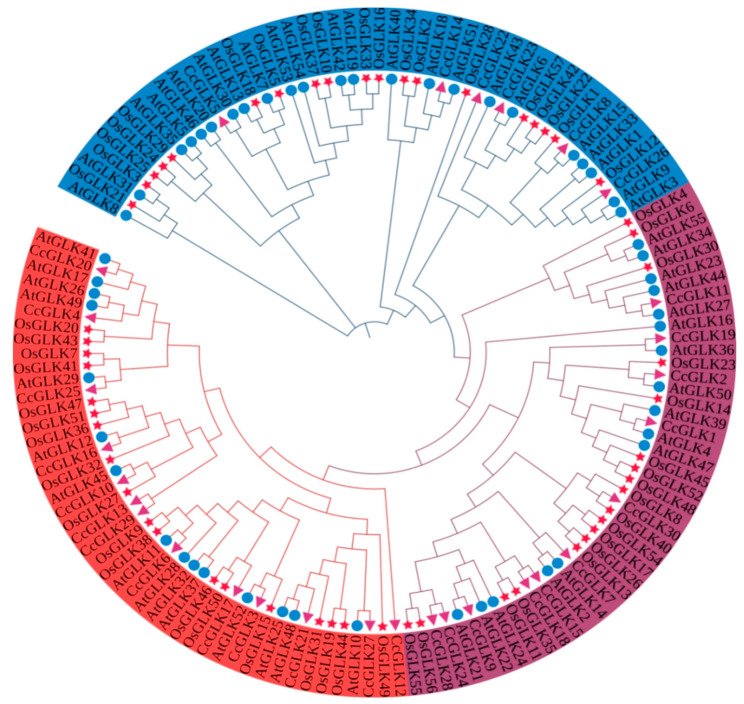
*GLK* gene phylogenetic tree based on protein sequences of three species. Blue represents Group I, purple represents Group II, and red represents Group III. Circles of different colors represent different subgroups. △ represents *Citrus*, ☆ represents *Oryza*, and ○ represents *Arabidopsis*.

**Figure 5 plants-13-00936-f005:**
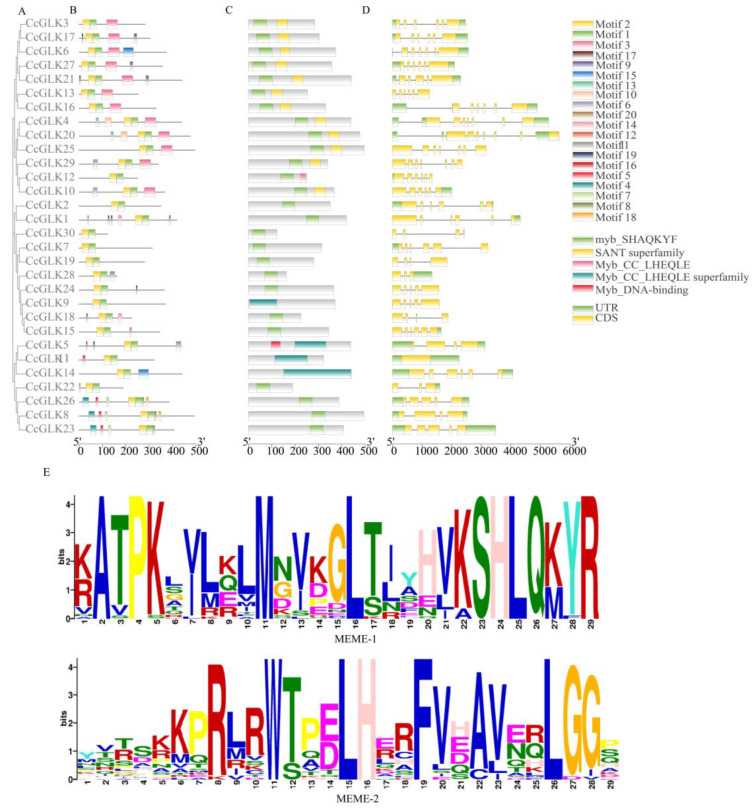
(**A**) Protein evolution analysis, (**B**) conservative motifs, (**C**) conservative domains, (**D**) gene structure, and (**E**) conservative domain sequences. Exons, introns, and Untranslated Regions (UTRs) are represented by yellow rounded rectangles, black lines, and green rounded rectangles, respectively.

**Figure 6 plants-13-00936-f006:**
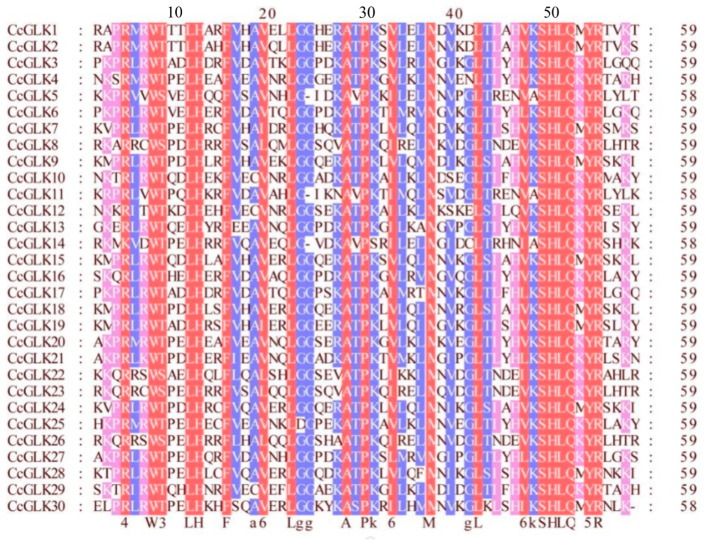
Multiple sequence alignment of *Citrus GLK’*s conservative domain. The same sequence is displayed in the same color.

**Figure 7 plants-13-00936-f007:**
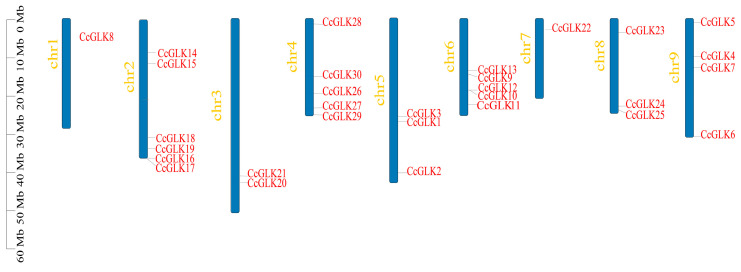
Chromosome positions.

**Figure 8 plants-13-00936-f008:**
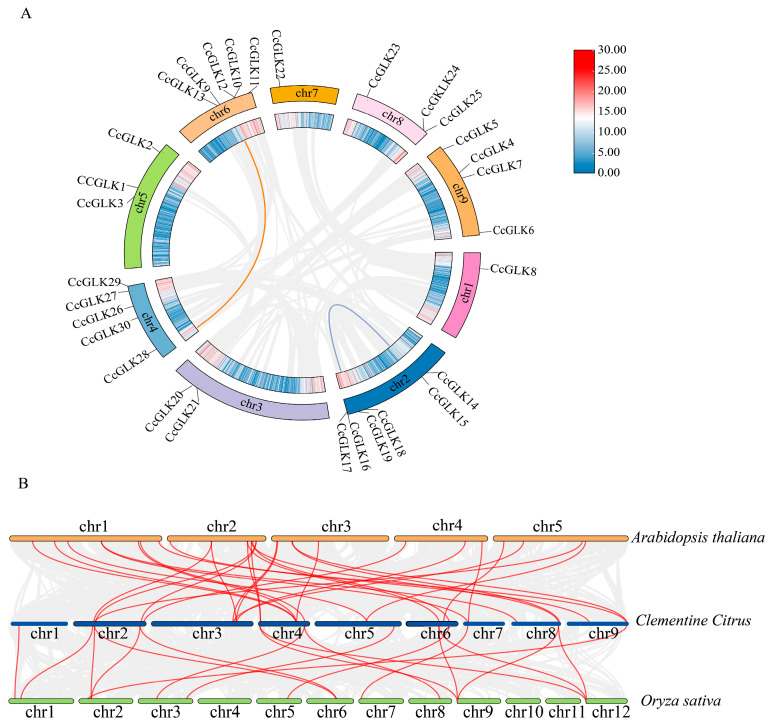
Collinearity analysis of *Citrus* GLK. (**A**) *GLK* gene replication event. The gray line represents all autosomes in the citrus genome, while the blue and red lines represent *CcGLK* gene replication events. (**B**) Collinearity analysis of *Citrus* and two model plants. The yellow chromosome represents *Arabidopsis*, the green chromosome represents rice, and the blue chromosome represents citrus.

**Figure 9 plants-13-00936-f009:**
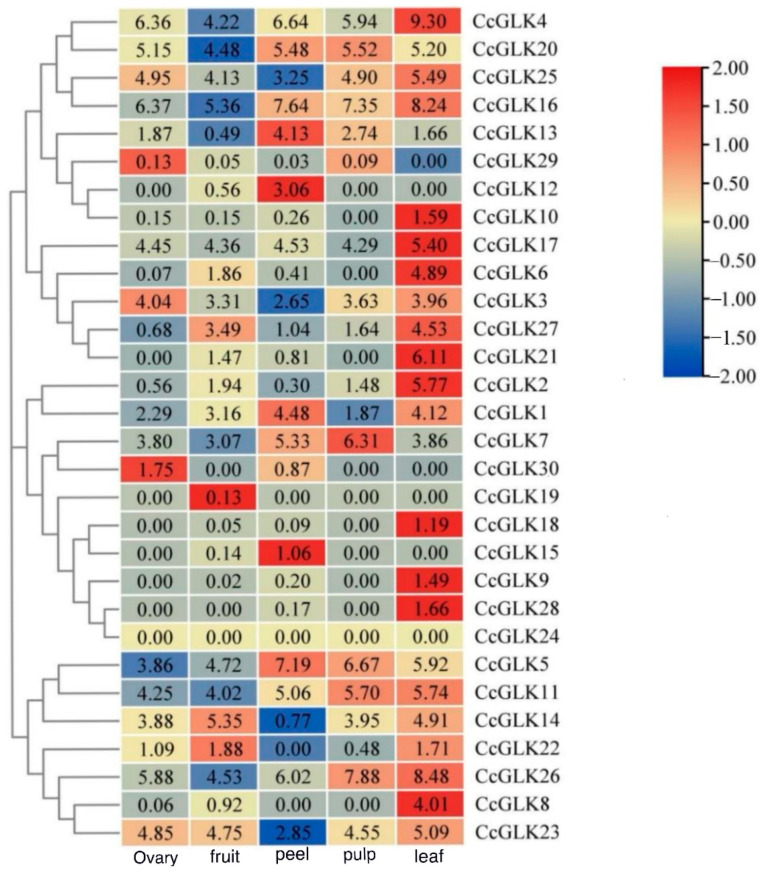
Expression patterns of *CcGLK*s in different tissues.

**Figure 10 plants-13-00936-f010:**
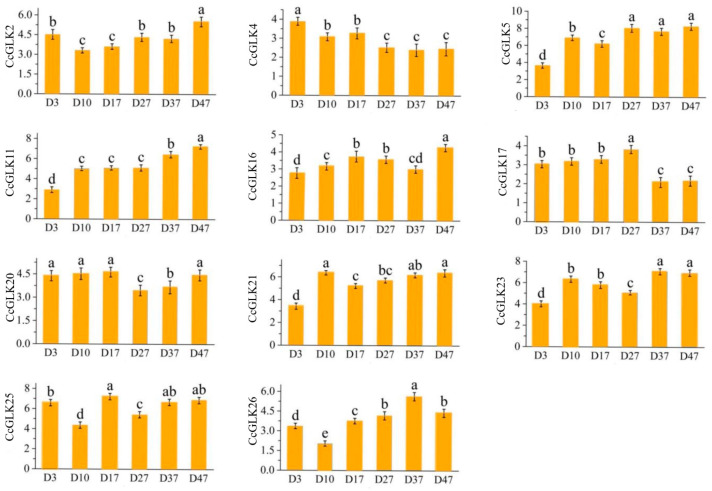
Relative expression levels of *CcGLK*s at different leaf growth stages. Different letters above the bar chart indicate significant differences between different treatments (*p* < 0.05).

**Figure 11 plants-13-00936-f011:**
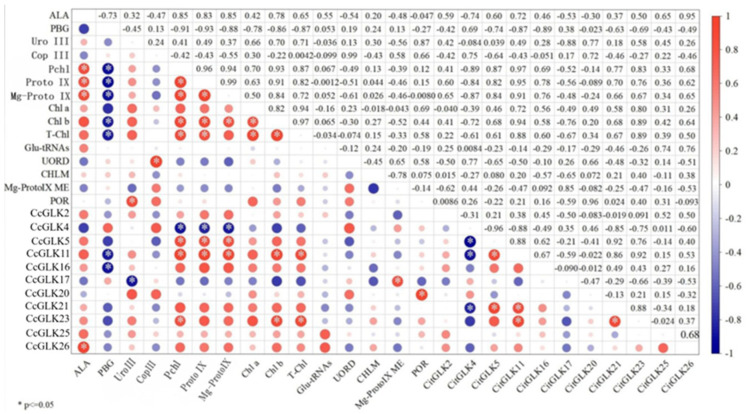
Correlation between *CcGLK*s and chlorophyll precursors and precursor synthases.

**Figure 12 plants-13-00936-f012:**
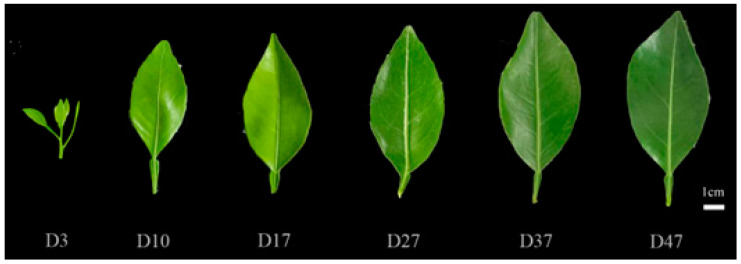
Leaf growth of ‘Kanpei’ at different leaf ages.

**Table 1 plants-13-00936-t001:** Physical and chemical properties of *CcGLK*s.

Group	Number of aa	MolWt	pI	II	GRAVY	Number of Introns
I	250–496	28,464.01–54,724.06	5.57–8.84	27.46–71.22	−0.972–−0.476	5–8
II	122–419	14,435.59–46,634.53	7.23–10.17	41.32–62.37	−1.057–−0.578	2–5
III	189–494	20,919.32–54,020.85	6.01–10.46	43.87–70.95	−1.032–−0.246	0–5

**Table 2 plants-13-00936-t002:** Ka and Ks values of collinear Cs*GLK* gene pairs.

Sequence 1	Sequence 2	Ka	Ks	Ka/Ks	Copy Type
*CcGLK15*	*CcGLK18*	0.326772	1.449596	0.225422539	tandem duplication
*CcGLK28*	*CcGLK9*	0.358406	1.893917	0.189240602	tandem duplication

**Table 3 plants-13-00936-t003:** *Arabidopsis–Citrus GLK* homologous pairs.

Chromosomes	*Arabidopsis* Thaliana		Location	*Citrus* C Lementina	Location
chr1	AT1G32240.1	==	chr5	CcGLK1	27065813…27070133
chr5	AT5G45580.2	==	chr5	CcGLK3	25793408…25795877
chr1	AT1G79430.2	==	chr9	CcGLK6	31030857…31033422
chr3	AT3G12730.1	==	chr9	CcGLK6	31030857…31033422
chr2	AT2G02060.2	==	chr9	CcGLK7	12119033…12122259
chr2	AT2G38300.1	==	chr6	CcGLK9	16820952…16822532
chr2	AT2G40970.1	==	chr6	CcGLK11	21920357…21922631
chr5	AT5G06800.2	==	chr6	CcGLK12	18766399…18767741
chr2	AT2G20570.2	==	chr2	CcGLK14	10399864…10403935
chr5	AT5G44190.1	==	chr2	CcGLK14	10399864…10403935
chr2	AT2G42660.1	==	chr2	CcGLK15	11280731…11282377
chr2	AT2G42660.1	==	chr2	CcGLK18	34240828…34242711
chr4	AT4G04555.1	==	chr2	CcGLK19	35451063…35452097
chr2	AT2G20400.2	==	chr3	CcGLK20	42990251…42995979
chr3	AT3G04450.1	==	chr3	CcGLK20	42990251…42995979
chr4	AT4G28610.1	==	chr3	CcGLK20	42990251…42995979
chr3	AT3G04030.3	==	chr3	CcGLK21	41238803…41241110
chr5	AT5G18240.4	==	chr3	CcGLK21	41238803…41241110
chr1	AT1G49560.1	==	chr7	CcGLK22	2881144…2882746
chr4	AT4G37180.2	==	chr7	CcGLK22	2881144…2882746
chr1	AT1G49560.1	==	chr8	CcGLK23	3625136…3628623
chr2	AT2G40260.1	==	chr8	CcGLK24	24682989…24684558
chr3	AT3G13040.1	==	chr8	CcGLK25	24976315…24979483
chr1	AT1G25550.1	==	chr4	CcGLK26	19605473…19608059
chr1	AT1G68670.1	==	chr4	CcGLK26	19605473…19608059
chr1	AT1G13300.1	==	chr4	CcGLK26	19605473…19608059
chr3	AT3G25790.1	==	chr4	CcGLK26	19605473…19608059
chr1	AT1G69580.2	==	chr4	CcGLK27	23957038…23959284
chr2	AT2G40260.1	==	chr4	CcGLK28	1461989…1463323
chr2	AT2G38300.1	==	chr4	CcGLK28	1461989…1463323

**Table 4 plants-13-00936-t004:** *Rice–Citrus GLK* homologous pairs.

Chromosomes	*Oryza sativa*		Location	*Citrus* Clementina	Location
chr1	LOC_Os01g08160.1	==	chr1	CcGLK8	5067104…5069631
chr1	LOC_Os01g13740.1	==	chr2	CcGLK14	10399864…10403935
chr2	LOC_Os02g14490.1	==	chr2	CcGLK18	34240828…34242711
chr2	LOC_Os02g14490.1	==	chr2	CcGLK15	11280731…11282377
chr6	LOC_Os06g35140.1	==	chr2	CcGLK18	34240828…34242711
chr6	LOC_Os06g35140.1	==	chr2	CcGLK15	11280731…11282377
chr11	LOC_Os11g01480.1	==	chr4	CcGLK28	1461989…1463323
chr12	LOC_Os12g01490.1	==	chr4	CcGLK28	1461989…1463323
chr3	LOC_Os03g20900.1	==	chr4	CcGLK27	23957038…23959284
chr8	LOC_Os08g33750.1	==	chr4	CcGLK27	23957038…23959284
chr3	LOC_Os03g55760.1	==	chr5	CcGLK2	40569723..40573126
chr11	LOC_Os11g01480.1	==	chr6	CcGLK9	16820952…16822532
chr12	LOC_Os12g01490.1	==	chr6	CcGLK9	16820952…16822532
chr5	LOC_Os05g34110.1	==	chr6	CcGLK11	21920357…21922631
chr7	LOC_Os07g02800.3	==	chr7	CcGLK22	2881144…2882746
chr11	LOC_Os11g01480.1	==	chr8	CcGLK24	24682989…24684558
chr12	LOC_Os12g01490.1	==	chr8	CcGLK24	24682989…24684558
chr2	LOC_Os02g07770.1	==	chr9	CcGLK6	31030857…31033422

**Table 5 plants-13-00936-t005:** The primer sequences of *CcGLK*s.

Gen Name	Forward Primer	Reverse Primer
*CcGLK2*	F: CCAGGGACACAGAGCAGAAC	R: TGGTGGAGGCTGTTGGTTTT
*CcGLK4*	F: CCCCGTAATCAAGGTAGT	R: GACCCTGTCTTCTGTAGCAC
*CcGLK5*	F: ATGACGAAGGCTCGCTGAAA	R: CTGGTGCTTCCCCTCCATTT
*CcGLK11*	F: CCGGCTACTGACCACTTGTT	R: ACAGGCACATACGGCAAGAA
*CcGLK16*	F: TGACTCCTCTTCGGATGGGAAA	R: TGCAGTCGTTTTTGCACCTC
*CcGLK17*	F: CGTCACTCAACTAGGCGGTC	R: GAGTCACCCATTTCTCTCCCC
*CcGLK20*	F: GGCCAGATTCATCAGAAGGGT	R: GCAAATTTCTTTGTATCTCAAGCTG
*CcGLK21*	F: GTCATCTGGTGTAGGTCCAGT	R: GCTCTTGTTTGGTTGGGGTC
*CcGLK23*	F: CCACCTCAGCAACAGACCAA	R: CTGCAAGGCACTGACAAACC
*CcGLK25*	F: GCAACCAGCCGAGTCAAAAA	R: AGGCGAATCCTTTTCAGGCA
*CcGLK26*	F: GGGGAAAAGAAGGACCAGCA	R: TGCAATTCAGGCGACCAAGA
*Actin*	F: CATCCCTCAGCACCTTCC	R: CCAACCTTAGCACTTCTCC

## Data Availability

All the data generated or analyzed during this study are included in this published article.

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
