# Peer review of "Genome-Wide Analysis of the GLK Gene Family and Its Expression at Different Leaf Ages in the Citrus Cultivar Kanpei"

_plants, 2024, doi:10.3390/plants13070936_

Round 1

Reviewer 1 Report

Comments and Suggestions for Authors

Dear Authors,

Thank you for submitting the manuscript. The work is pertinent in the current context but it requires more vigor.

1. The phylogenetic tree needs bootstrap values to be added. Wiothout this the tree is unfit.

2. Can the secondary structure prediction be added in the MSA analysis.

3. Please add the localization of the genes identified in the table.

4. Can the PCA analysis be incorporated with all the traits studied from the corelation analysis.

5. Need a predictive model for the discovery as in how the precusrors or the genes are going to effectively treat the plant at pulp et level.

6. In the stages of the leaf figure mention the approx no of days old.

7. Material mnethod is not written very effectively. Please elaborate cite and improve it.

6. 

Reviewer 2 Report

Comments and Suggestions for Authors

A total of thirty CcGLK transcription factors 14 (TFs) were discovered in the citrus genome, distributed across all nine chromosomes. The 15 low occurrence of gene tandem duplication events and intronic variability suggests that 16 intronic variation may be the primary mode of evolution for CcGLK TFs. Tissue-specific 17 expression patterns were observed for various GLK family members;

The manuscript is relatively completed, the results are reliable, and the manuscript can be published;

It is proposed that tables 4 and 5 be merged;

There are some a little places in it that pay attention to capitalization, Latin name italics, etc.

Comments on the Quality of English Language

I think the English is OK

Round 2

Reviewer 1 Report

Comments and Suggestions for Authors

Dear Authors,

Congratulations on improving the work and content. It is good for publication. No further changes required.
